# Role of Probiotics in Modulating Human Gut Microbiota Populations and Activities in Patients with Colorectal Cancer—A Systematic Review of Clinical Trials

**DOI:** 10.3390/nu13041160

**Published:** 2021-04-01

**Authors:** Adrianna Wierzbicka, Dorota Mańkowska-Wierzbicka, Marcin Mardas, Marta Stelmach-Mardas

**Affiliations:** 1Department of Obesity Treatment, Metabolic Disorders and Clinical Dietetics Poznań University of Medical Sciences, Szamarzewskiego 84, 60-569 Poznań, Poland; duniakaw@gmail.com; 2Department of Gastroenterology, Metabolic Diseases, Internal Medicine and Dietetics, Poznan University of Medical Sciences, 84, 60-569 Poznań, Poland; dmankowska.wierzbicka@gmail.com; 3Department of Oncology, Poznan University of Medical Sciences, 84, 60-569 Poznań, Poland; marcin.mardas@ump.edu.pl

**Keywords:** probiotics, gut microbiota, human microbiome, colorectal cancer, chronic diseases, micronutrients, supplementation, disease prevention

## Abstract

Background: Growing attention has been given to the role of nutrition and alterations of microbial diversity of the gut microbiota in colorectal cancer (CRC) pathogenesis. It has been suggested that probiotics and synbiotics modulate enteric microbiota and therefore may be used as an intervention to reduce the risk of CRC. The aim of this study was to evaluate the influence of probiotics/synbiotics administration on gut microbiota in patients with CRC. Methods: PubMed, Scopus, and Web of Science were searched between December 2020 and January 2021. Randomized controlled trials (RCTs) recruiting adults with CRC, who have taken probiotics/synbiotics for at least 6 days were included. Changes in gut microbiota and selected biochemical and inflammatory parameters (i.e., hsCRP, IL-2, hemoglobin) were retrieved. Results: The search resulted in 198 original research articles and a final 6 were selected as being eligible, including 457 subjects. The median age of patients was 65.4 years old and they were characterized by the median BMI value: 23.8 kg/m^2^. The literature search revealed that probiotic/synbiotic administration improved enteric microbiota by increasing the abundance of beneficial bacteria such as *Lactobacillus, Eubacterium, Peptostreptococcus, Bacillus* and *Bifidobacterium*, and decreased the abundance of potentially harmful bacteria such as *Fusobacterium, Porhyromonas, Pseudomonas* and *Enterococcus*. Additionally, probiotic/synbiotic intervention improved release of antimicrobials, intestinal permeability, tight junction function in CRC patients. Conclusions: The use of probiotics/synbiotics positively modulates enteric microbiota, improves postoperative outcomes, gut barrier function and reduces inflammatory parameters in patients suffering from CRC.

## 1. Introduction

In 2020, an estimated 147,950 new cases and an estimated 53,200 deaths will be attributed to Colorectal cancer (CRC) [1]. Given the multifactorial etiology, CRC is associated with nutrition, inflammatory processes, and genetic factors [2,3,4]. Based on the evidence, the consumption of processed meat, alcoholic beverages, and the accumulation of body fat significantly increases the risk of the development of the disease [4,5], whereas the consumption of fiber, calcium, milk, whole grains, vegetables and fruit reduces the risk of CRC [6]. Recently, growing attention has been given to the role of gut microbiota (i.e., *Fusobacterium nucleatum, Escherichia coli*, *Enterococcus faecalis*, *Streptococcus gallolyticus, Bacteroides fragilis*) in colorectal carcinogenesis [7]. The microbial diversity of enteric bacteria occurs in the cancer tissues which are directly exposed to microbes, such as the colon and rectum. Dysbiosis can be defined as an increase in proinflammatory species and a decrease in microbial diversity. Thus, any microbial imbalance may produce carcinogenic and genotoxic metabolites as well as trigger inflammatory process [8]. It has been shown that the supplementation of probiotics might be required to re-establish the homeostasis and to restore the environment [9,10]. Furthermore, probiotics benefit a host through several physiological functions such as protecting against pathogens, regulating host immunity, and strengthening the gut integrity [11,12,13]. In the context of CRC patients, the administration of probiotics may protect them from treatment-associated side effects [14,15]. Currently, probiotic supplementation has been proven to be a promising innovative approach to counteracting CRC progression [16]. In particular, some bacteria, mainly of the genera *Lactobacillus* and *Bifidobacterium*, have a potentially beneficial role in modulating the anti-inflammatory response and protecting against enteric pathogens [17,18]. To date, their impact on gut homeostasis has been widely investigated.

Therefore, the aim of the study was to evaluate the influence of probiotics/synbiot-ics administration on gut microbiota changes in patients with CRC.

## 2. Materials and Methods

### 2.1. Search Strategy and Study Selection

The study was performed according to guidelines of the Preferred Reporting Items for Systematic Reviews and Meta-Analysis (PRISMA) [19].

A comprehensive search was conducted using the electronic databases: PubMed, Scopus, and Web of Science between December 2020 and January 2021 to identify randomized, controlled trials (RCT), clinical trials (CT), double-blind placebo trials (DBPT) run in adults (age over 18) diagnosed with either colorectal or colon or rectal cancer where an intervention with probiotics/symbiotic was analyzed in the context of gut microbiota changes. Studies recruiting adults with CRC who have taken probiotics/synbiotics for at least 6 days were included. Briefly, probiotics were defined according to the Food and Agricultural Organization of the United Nations and the World Health Organization as: “live microorganisms, which, when consumed in adequate amounts, confer a health benefit on the host” [20] and synbiotics as “mixtures of probiotics and prebiotics that beneficially affect the host by improving the survival and implantation of live microbial dietary supplements in the gastrointestinal tract of the host” [21].

Search terms included: (“Colorectal cancer“ OR “Colonic Neoplasms” OR “Adenomatous Polyposis Coli” OR “Colorectal Neoplasms, Hereditary Nonpolyposis” OR “Rectal Neoplasms” OR “Colorectal Carcinoma” OR “Sigmoid neoplasms” OR “Anus Neoplasms” OR “Anal Gland neoplasms) AND (“Probiotic *” OR “Lactobacillus”, ”Bacillus coagulans“ OR “Propionibacterium” OR “Bifidobacterium” OR “Saccharomyces”) AND (“Microbiota” OR “Human Microbiome” OR “ Microbial Community” OR “ Microbial Community Composition” OR “Microbial “Community Structure” OR “Microbiome” OR “Gastrointestinal Microbiome” OR “Microbial Consortia” OR “Metagenome” OR “Mycobiome” OR “Periphyton” OR “Gut Microbiota” OR “Gut Microbiome”) were assessed.

The search was restricted to studies published in the English language and carried only in humans. Exclusion criteria included: conference publications, articles available only in abstract form (no possible contact with authors).

### 2.2. Data Extraction

Identified studies from the electronic databases were screened by title and abstract by two independent reviewers. Full texts were retrieved if decisions could not be made based on the information provided in the abstract. Disagreements regarding selection were resolved by discussion or consensus by all authors. After determination of study selection, the following data were extracted: first author’s name, year of publication, study design, sample size, mean age and gender of trial participants, Body Mass Index (BMI), microbiota composition changes, method used for bacterial DNA isolation, probiotic strain, dosage in grams and duration of probiotic intervention. Additionally, the stage of colon cancer and its location were collected. Changes in selected biochemical and inflammatory parameters (i.e., hsCRP, IL-2, hemoglobin) were retrieved.

### 2.3. Risk of Bias Tool

In order to assess the methodological quality of the studies included in this review, we used the “Cochrane Handbook of Systematic Reviews of Interventions” [22], evaluating the risk of bias in each of the proposed items: (a) selection bias, (b) performance bias, (c) detection bias, (d) attrition bias and (e) reporting bias.

## 3. Results

### 3.1. Study Selection and Characteristics of Study Population

The detailed steps of the study selection process are given as a flowchart in Figure 1. An initial literature search generated a total of 198 articles. After removing duplicates, the titles and abstracts were screened independently by two reviewers. By evaluating the titles and abstracts and full-text, additional studies were excluded as irrelevant (unrelated topics or irrelevant design). Finally, 6 studies [23,24,25,26,27,28] with a total of 457 participants were included in the study. All of them were RCTs with a duration ranging up to 78 days [27], published between 2007–2017. The median age of patients was 65.4 years old and they were characterized by the median BMI value: 23.8 kg/m^2^. Most of the studies were conducted in the European population [26,27,28], two trials in Japanese [23,24], and one in Chinese [25]. Only one study mentioned diet, where patients received regular diet preoperatively and a low residue diet one day before the surgery [25]. Colorectal cancer staging was performed in two studies using the Classification of Malignant Tumors (TNM), where T refers to the size and extent of the main tumor, N to the number of nearby lymph nodes that have malignant changes and M refers to distal metastasis [24,27] and one study using the Astler–Coller classification [25]. Tumor location was assessed in four studies [23,24,25,26]. In the majority of patients the tumor was localized in the colon. The following probiotics strains were used: *Bifidobacterium longum* BB536 [23]; *Enterococcus faecalis* T110, *Clostridium butyricum* TO-A *Bacillus mesentericus TO-A* [24]; *Bifidobacterium lactis* Bl-04 (ATCC SD5219), *Lactobacillus acidophilus* NCFM (ATCC 70039) [27]; *Lactobacillus plantarum* (CGMCC No. 1258), *Lactobacillus acidophilus* LA-11, *Bifidobacterium longum* BL-88 [25]; *Bifidobacterium longum* (BB536), *Lactobacillus johnsonii* (La1) [26]; Oligofructose enriched inulin (SYN1), *Lactobacillus rhamnosus* GG (LGG), and *Bifidobacterium lactis* Bb12 (BB12) [28]. Table 1 and Table 2 present the detailed characteristics of the selected studies.

### 3.2. Effect of Probiotics on Changes in Gut Microbiota and Postoperative Outcome

Alterations in the enteric microbiota were reported in all studies [23,24,25,26,27,28]. We summarized the representative taxa at two levels (phylum and genus levels) and the effects of the probiotic intervention on the fecal flora of cancer patients were presented in Table 3. In five studies, microbiota was evaluated by means of the PCR amplification technique [23,24,25,26,27]. In one study standard plate count techniques were used [27]. The supplementation with a combination of *Lactobacillus bacteria*, including *L. acidophilus*, *L. plantarum*, *L. rhamnosus* species and *Bifidobacterium* bacteria [24,26,27], including *B. lactis* and *B. longum* species was associated with an increased abundance of positive bacteria such as: *Bifidobacterium*, *Lactobacillus*, *Bacillus*, *Eubacterium* and *Peptostreptococcus*. Moreover, the use of this probiotic strain mix reduced the enteropathogenic bacteria: *Enterococcus*, *Fusobacterium*, *Porphyromonas* and *Pseudomonas* [24,25,26,27]. Furthermore, intervention with *Enterococcus faecalis*, *Clostridium butyriucm* and *Bacillus mesentericus* increased the abundance of positive bacteria—*Bifidobacterium* [23].

Additionally, probiotic strains—*Bifidobacterium longum*, *Bifidobacterium lactis*, *Lactobacillus rhamnosus* and *Lactobacillus johnsoni*—were associated with positive changes in the biochemical and inflammatory parameters. *Bifidobacterium longum* supplementation increased serum hemoglobin, erythrocyte, lymphocyte, total protein and albumin concentration, whereas it decreased high sensitive C-reactive proteins (hsCRP) [23]. Furthermore, a combination of *Bifidobacterium longum* and *Lactobacillus acidophilus* reduced the expression of the dendritic phenotypes CD83-123, CD83-HLADR, and CD83-11c (markers of activation) [26]. Synbiotic administration (*Lactobacillus rhamnosus*, *Bifidobacterium lactis* and Oligofructose enriched inulin) increased barrier function and production of interferon-γ (IFN-γ) and decreased secretion of interleukin 2 (IL-2) [28].

The intervention with bacterial strains of *Lactobacillus plantarum, Lactobacillus acidophilus*, and *Bifidobacterium longum* [25] was associated with a lower postoperative incidence of bacterial translocation and increased mean colon mucosal transepithelial resistance. On the contrary, the use of these bacteria was associated with reduced transmucosal permeation of the lactulose/mannitol ratio, horseradish peroxidase and decreased ileal-bile acid-binding protein [25]. Furthermore, pre- and postsurgical supplementation with *Lactobacillus plantarum, Lactobacillus acidophilus*, and *Bifidobacterium longum* resulted in reduced hospitalization time and improved peristalsis of the intestine [23,25]. In addition, reduced postoperative abdominal distension, abdominal cramping and reduced pyrexia were observed [25]. Bacterial strains of *Enterococcus faecalis, Clostridium butyricum* and *Bacillus mesentericus* reduced postoperative and surgical superficial incisional infections and reduced the length of time prior to the passage of gas [24].

### 3.3. Risk of Bias Assessment

The distribution of biases classified as “low risk” or “unclear risk” was similar, except for detection bias, which presented a 80% “unclear” risk. Detection bias is related to the blinding of the assessors to the study results, and although the Cochrane manual states that their blinding does not ensure success, a lack of blinding could bias the study results. Nevertheless, the predominant classification of this type of bias, in particular, was “unclear risk” and not “high risk”, which is associated with a lack of information regarding this bias on the part of the authors, rather than a possible bias in the results (Figure 2).

## 4. Discussion

A number of studies have shown that the consumption of certain probiotics is able to beneficially alter the predisposing factors of CRC, and is a promising approach for management of CRC. Our study confirmed that particular combinations of strains or specific species of pro-/synbiotics have beneficial effects in CRC patients in terms of increasing antimicrobial defense, improving intestinal integrity and immune response. The most commonly investigated probiotic bacteria were *Lactobacillus* and *Bifidobacterium* [23,25,26,27,28]. The study presented by Hibberd et al. [27] suggested that intervention with probiotic mix of *Bifidobacterium lactis* Bl-04, *Lactobacillus acidophilus* NCFM markedly improved the abundance of butyrate-producing bacteria such as *Clostridiales* and *Faecalibacterium* species. The main products of the substrate fermentation in the gut are short-chain fatty acids (SCFAs) that interact with the intestinal microbiota and the host cell [29]. Additionally, the SCFA butyrate is responsible for the epithelial function through the induction of genes encoding tight junctions components. Due to their potential therapeutic anti-inflammatory components, some strains of *Lactobacilli* and *Bacillus* may positively influence the activity of inflammation [30]. Moreover, *Lactobacillus* and *Bifidobacterium* spp. may change the expression of genes involved in cell proliferation, apoptosis, cell death and metastasis [31]. A study conducted by Moore and Moore [32] showed that an increased abundance of *Eubacterium aerofaciens* and *Lactobacillus* was protective against the formation of colon polyps. Furthermore, reduced levels of the tumor-inducing microbial agents *Fusobacterium* and *Peptostreptococcus* were observed [27]. It has been reported that *Fusobacterium* spp. may possess virulence characteristics which contribute to the increased adhesiveness to host epithelial cells [33]. Finally, the overexpression of both *Fusobacterium* and *Peptostreptococcus* genera may promote proinflammatory environment and cause periodontitis [34]. Interestingly, *Pseudomonas* enhance the formation of a local anaerobic environment which is viable for colonization by *Peptostreptococcus, Fusobacterium* and *Lactococcus* [35]. He et al. [36] investigated the effect of probiotic/synbiotic supplementation in CRC patients and observed an improvement in enteric microbiota by increasing *Lactobacillus* and reducing the *Enterobacteriaceae* members. This stays in line with our study. Huycke et al. [37] showed that *Enterococcus faecalis* produces DNA damaging superoxide radicals and extracellular genotoxins which may contribute to CRC development. Another worthwhile study investigated the effect of probiotic mix (*B. longum, L. acidophilus,* and *E. faecalis*) on CRC patients who have undergo radical colorectomy [38]. A significant reduction in *Fusobacterium* species and improved diversity of the mucosa-associated microbiota were visible. Finally, this study suggested that probiotic supplementation may improve state of health of CRC by positive regulation and alteration of mucosal-associated microbiota [38]. In animal models the intervention with *Lactobacillus salivarius* Ren resulted in reduction in *Bacteroides dorei, Clostridiales, Ruminococcus* species and the level of *Prevotella* species increased [39].

Multiple studies have shown that by reducing intestinal permeability and increasing microbial diversity, probiotics are able to lower the rates of postoperative complications [17,18,19,36]. It has commonly been assumed that multiple factors such as enteric barrier disruption, increased intestinal permeability, and host immunologic compromise account for postoperative infectious complications [33,34]. One of the mechanisms of action of probiotics is the suppression of pathogens and the manipulation of gut microbiota by inducing the host’s secretion of IgA from plasma cells and β-defensins from intestinal epithelial cells [40]. *Lactobacillus rhamnosus* GG and *Bifidobacterium lactis* Bb-12 have been shown to enhance IgA production in the gut mucosa [41]. Furthermore, probiotics will secure the intestinal barrier through the mediation of cytokine secretion, the activation of natural killer cells, and as a result, will contribute to dendritic cell maturation [42]. *Lactobacillus gasseri* 4M13 and *Lactobacillus rhamnosus* 4B15 have shown anti-inflammatory properties by inhibiting the expression of inflammatory cytokines at transcriptional level in vitro [43]. In addition, certain probiotics, such as *Bifidobacterium longum, Lactobacillus acidophilus* ATCC, benefit the host by reducing intestinal inflammation by inhibiting the activation of nuclear factor κ B (NF- κB) [44,45,46]. *Lactobacilli*, through the production of bacteriocins, hydrogen peroxide and lactic acid, are able to inhibit intestinal pathogens [47]. Changes in the above pathways might also affect the proliferation and survival of target cells [40].

Recent scientific studies shed light upon the relationship between probiotic use and postoperative outcomes. Postsurgical complications such as diarrhea or pyrexia markedly increase the risk of infections and prolong hospital stay in patients with CRC [48]. It has been shown that supplementation with combination of *Lactobacillus* and *Bifidobacterium* bacteria resulted in significantly reduced hospitalization time and improved peristalsis of the intestine [23,25]. Furthermore, this systematic literature review (SLR) reported that probiotic administration contributed to reduced postoperative abdominal distension and abdominal cramping. Similarly, Tan et al. [49] studied the influence of preoperative probiotic mixture (*Lactobacillus acidophilus, Lactobacillus casei, Lactobacillus lactis, Bifiobacterium infantis, Bifiobacterium bifidum,* and *Bifiobacterium longum)* on patients with CRC. A markedly reduced hospitalization period and time needed for regaining normal enteric function after surgery were observed in the intervention group. Previous studies [50,51] confirmed that the probiotic/synbiotic intervention among antibiotic therapy significantly reduced hospitalization time and septic complications. This is critically important since shorter antibiotic treatment results in a decrease in emerging bacterial resistance. Furthermore, intervention with *B. longum*, *L. acidophilus*, and *E. faecalis* reduced the prevalence of diarrhea and significantly improved the bowel movement in CRC patients [52]. This SLR revealed that perioperative probiotic supplementation of *Enterococcus faecalis* T110, *Clostridium butyricum* TO-A, and *Bacillus mesentericus* TO-A significantly reduced postoperative superficial incisional surgical site infections [24]. In line with these observations, the administration of *Lactobacillus casei* along with *Bifidobacterium breve* markedly reduced the incidence of postsurgical infectious complications [53].

Moreover, an anti-inflammatory response was visible within this SLR, as described by the significant reduction in serum hsCRP, IL-2, CD83-123, CD83-HLADR and CD83-11c concentrations interlinked with significant increase serum concentrations of IFN- γ, CD3, CD4, CD8, hemoglobin, erythrocytes, lymphocytes, total protein [23,26,28]. Actinobacteria were positively associated with erythrocytes, hemoglobin, albumin, NK cell activity, whereas *Firmicutes* were positively associated with albumin, total protein, and lymphocytes, but negatively correlated with hCRP, IL-6 and white blood cells [23]. On contrary, supplementation with *Lactobacillus plantarum* BFE 1685 and *Lactobacillus johnsonii* BFE 6128 have been proven to help in modulating the immune system by inducing the release of the cytokine IL-8 in vitro [54]. Furthermore, the supplementation of *Lactobacillus acidophilus, Lactobacillus lactis, Lactobacillus casei and Bifidobacterium longum, Bifidobacterium bifidum, Bifidobacterium* has been proven to significantly reduce the level of proinflammatory cytokines, IL-6, IL-10, IL-12, IL-17A, IL-17C, IL-22 and TNF-alpha in intervention group comparing to placebo [55]. In animal models the effect of *Lactobacillus plantarum* and *Lactobacillus rhamnosus* strains was investigated where the supplementation with *Lactobacillus plantarum* markedly increased the lifespan of tumor-bearing mice and diminished the CT26 cell growth by enhancing the CD8+ function, Th1-type CD4 + T differentiation, IFN-*γ* expression and NK cell infiltration compared to *Lactobacillus rhamnosus* [56].

There are some limitations to this study. Firstly, all the included studies were published only in English, which might lead to the loss of valuable studies in different languages. Secondly, the retrieved studies were marred by significant heterogeneity including the strain, amount, duration, and schedule of the pro-/symbiotics intervention. Thirdly, there was a lack of uniformity in the accompanying medications and surgical intervention in each of the included studies. Finally, the detection bias was related to the blinding of the assessors to the study results and could have an influence on the study results.

## 5. Conclusions

In conclusion, current review indicated that the use of pro-/synbiotics may have a positive influence on enteric microbiota and gut barrier function, which may be related to the improvement in postoperative outcomes. Furthermore, amongst combination probiotics, and the nine-strain combination of two *Bifidobacterium,* one *Enterococcus,* one *Bacillus,* one *Clostridium,* four *Lactobacillus* was associated with improvement in overall symptoms. Moreover, the increased abundance of beneficial bacteria such as *Lactobacillus* and *Bifidobacterium* was observed during intervention. Nevertheless, still there is a need to prove probiotic administration in CRC patients before we consider this approach as promising for the treatment of CRC.

## Figures and Tables

**Figure 1 nutrients-13-01160-f001:**
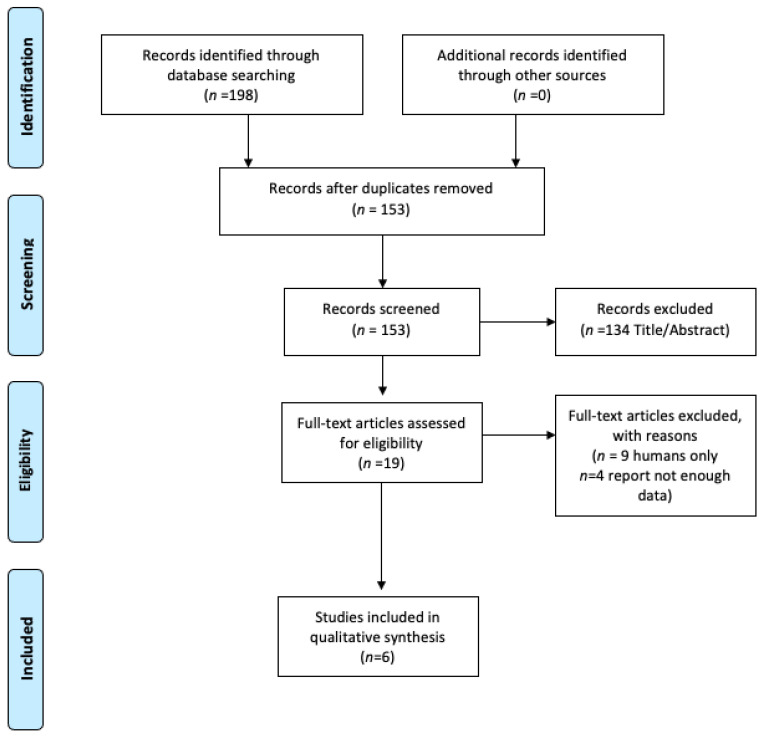
Flow chart of the databases search on influence of probiotics/synbiotics administration on gut microbiota in patients with colorectal cancer (CRC).

**Figure 2 nutrients-13-01160-f002:**
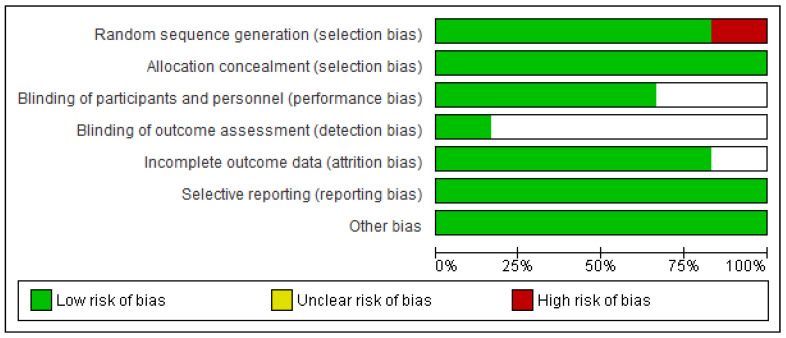
Risk of bias graph: review authors′ judgements about each risk of bias item presented as percentages across all included studies.

**Table 1 nutrients-13-01160-t001:** Characteristics of enrolled population (*n* = 457).

Source, Year	Trial Type	Sample Size (*n*)			Age (years)			Sex (M/F *n*)			BMI (kg/cm^2^)			Stage		Location		
		Control Group	Probiotic Group		Control Group	Probiotic Group		Control Group	Probiotic Group		Control Group	Probiotic Group		Control Group	Probiotic Group	Control Group	Probiotic Group	
**Mizuta et al., 2016**	PRCL	*n* = 29	*n* = 31		71.2 ± 9.5	68.9 ± 10.4		15/14	20/11		24.1 ± 3.4	22.4 ± 3.7		n/A	n/A	Colon 12 Rectum 13 Others 4 ***	Colon 11 Rectum 19 Others 0	
**Aisu et al., 2014**	RCT	*n* = 81	*n* = 75		69.1 ± 11.3	68.0 ± 13.8		44/37	47/28		23.3 ± 3.8	21.7 ± 2.7		I 29 II 32 IIIA 11 IIIB 3 IV 6 *	I 31 II 16 IIIA 10 IIIB 3 IV 8 *	Colon 3 Ascending colon 8 Transverse colon 6 Descending colon 3 Sigmoid colon 25 Rectum 22	Colon 5 Ascending colon 14 Transverse colon 5 Descending colon 0 Sigmoid colon 16 Rectum 8	
**Hibberd et al., 2017**	RCT	*n* = 21	*n* = 15		63 (55–73)	77 (68–75)		4/17	6/9		n/A	24.1 (22.5–24.8)		n/A	I 2 II 6 III 7 *	n/A	n/A	
**Liu et al., 2010**	RDBT	*n* = 50	*n* = 50		65.7 ± 9.9	65.3 ± 11.0		31/19	28/22		22.6 ± 2.0	22.8 ± 1.8		A 12 B 29 C 9 **	A 11 B 30 C 9 **	Transverse Colon 8 Descending Colon 10 Sigmoid colon 21 Rectum 11	Transverse Colon 7 Descending Colon 5 Sigmoid colon 25 Rectum 13	
**Gianotti et al., 2010**	RDBT	*n* = 10	Low dose	High dose	63.3 ± 10.2	Low dose	High dose	7/3	Low dose	High dose	25.6 ± 2.6	Low dose	High dose	n/A	n/A	Left colon 4 Right colon 3 Rectum 3	Low dose	High dose
			*n* = 11	*n* = 10		64.7 ± 4.8	62.7 ± 7.8		8/3	7/3		26.5 ± 4.1	24.4 ± 3.7				Left colon 6 Right colon 2 Rectum 3	Left colon 5 Right colon 2 Rectum 3
**Rafter et al., 2007**	RDBPCT	*n* = 40	*n* = 34		57.0 ± 9.75	61.1 ± 5.55		22/18	21/13		n/A	n/A		n/A	n/A	n/A	n/A	

PRCL—Prospective randomized clinical trial, RCT—randomized control trial, RDBT—randomized double-blind trial, RDBPCT—randomized, double-blind placebo controlled * TNM staging system: 0—no evidence of cancer in the colon or rectum; I—tumor has grown into the submucosa; II—tumor has grown into the muscularis propria; IIIA—cancer has grown through the inner lining or into the muscle layers of the intestine. It has spread to 1 to 3 lymph nodes or to a nodule of tumor in tissues around the colon or rectum that do not appear to be lymph nodes but has not spread to other parts of the body; IIIB—the cancer has grown through the bowel wall or to surrounding organs and into 1 to 3 lymph nodes or to a nodule of tumor in tissues around the colon or rectum that do not appear to be lymph nodes. It has not spread to other parts of the body; IV—tumor has grown into the surface of the visceral peritoneum ** Astler–Coller classification: A—limited to mucosa; B—extending into/penetrating through muscularis propria, nodes not involved; C—extending into/penetrating through muscularis propria, nodes involved.

**Table 2 nutrients-13-01160-t002:** Probiotic supplementation and CRC: outcomes of clinical studies.

Scheme	Probiotic Intervention Dose (g)	Duration (Days)	Key Results
**Mizuta et al., 2016**	2 g of *Bifidobacterium longum* BB536 powder (approximately 5 × 10^10^ CFU/2 g)	21–28	↑ anti-inflammatory response(↓ high sensitive C-reactive proteins, ↑ postoperative levels of erythrocytes, hemoglobin, lymphocytes, total protein, and albumin) ↓ duration of hospital stay
**Aisu et al., 2014**	2 mg *Enterococcus faecalis* T110, 0.01 g *Clostridium butyricum* TO-A and 0.01 g *Bacillus mesentericus* TO-A 6 × 10^9^ CFU/d	15	↓ incidence of postoperative complications (↓ time of flatus, ↓ time of meal intake, ↓ superficial incisional infections
**Hibberd et al., 2017**	1.4 × 10^1^⁰ CFUs *Bifidobacterium lactis* Bl-04 (ATCC SD5219), 7 × 10⁹ CFUs *Lactobacillus acidophilus* NCFM (ATCC 700396) and 0.63 g inulin.	8–78	↑ anti-inflammatory response ↑ microbial diversity: α- diversity and β-diversity
**Liu et al., 2010**	*Lactobacillus plantarum* (CGMCC No. 1258, cell count ≥ 10^11^ CFU/g), *Lactobacillus acidophilus* (LA-11, cell count ≥ 7.0 × 10^1^⁰ CFU/g) and *Bifidobacterium longum* (BL-88, cell count ≥ 5.0 × 10^1^⁰ CFU⁄ g)	16	↓ incidence of postoperative complications (↓ abdominal cramping, ↓ abdominal distention, ↓ duration of pyrexia ↓ time to first defecation) ↓ incidence of diarrhea ↑ microbial diversity: α- diversity and β-diversity
**Gianotti et al., 2010**	2 × 10⁷ CFU/d of a mixture of *Bifidobacterium longum* (BB536) and *Lactobacillus johnsonii* (La1)	6	↑ anti-inflammatory response (↑ CD3, CD4, CD8, dendritic phenotypes CD83-123, ↓ CD83-HLA DR, CD83-11c)
**Rafter et al., 2007**	Oligofructose enriched inulin (SYN1) and *Lactobacillus rhamnosus* GG (LGG) and *Bifidobacterium lactis* Bb12 (BB12), 12 g SYN1 together with the probiotic capsule > log_10_ CFU/g	42	↑ anti-inflammatory response (↑ interferon γ, ↓ interleukin (IL) 2) ↓ proliferation rate of colorectal cells

CFU—colony-forming units.

**Table 3 nutrients-13-01160-t003:** Effect of probiotics on specific bacteria.

Source, Year	Bacteria		Effect of Probiotic	Probiotic Group (Mean ± SD)				Control Group (Mean ± SD)		Method Used for Bacterial DNA Isolation
	Phylum	Genus		Before		After		Before	After	
**Mizuta et al., 2016**	Actinobacteria	n/A	Increase	0.24–1.90		0.36–3.09 ***		0.32–4.89	0.21–2.60	PCR amplificationof the V3-V4 region of bacterial 16S rRNA genes obtained from fecal samples
	Bacteroidetes	n/A	No change	18.88–32.89		24.76–32.87		18.32–32.01	27.17–40.60 ***	
	Firmicutes	n/A	Decrease	52.34–72.98		48.46–64.15 ***		57.18–75.96	46.77–64.24	
	Proteobacteria	n/A	No change	1.54–5.06		2.27–9.75 ***		1.50–2.16	2.90–5.84	
**Aisu et al., 2014**	Actinobacteria	Bifidobacterium *	Increase	4.6%		9.1% ***		7%	5.8%	PCR amplification of the 16S rDNA genes obtained from fecal samples
**Hibberd et al., 2017**	Firmicutes Firmicutes	Eubacterium * Peptostreptococcus *	Increase Increase	n/A 2.1 ± 2.6		2.9 ± 2.7 *** 0.04 ± 0.06 ***		n/A 0.00 ± 0.00	0.86 ± 1.8 0.42 ± 0.71 ***	PCR amplification of the V4 variable region of the 16S rRNA gene obtained from mucosa and tumor tissue as well as from fecal samples
	Fusobacteria Bacteroidetes	Fusobacterium ** Porphyromonas **	Increase Increase	7.6 ± 7.8 n/A		0.03 ± 0.05 *** 0.00 ± 0.00		0.23 ± 0.60 n/A	0.81 ± 0.87 0.43 ± 0.56 ***	
	Faecalibacterium Firmicutes Firmicutes Actinobacteria	n/A	Decrease	n/A		6.5 ± 2.0 ***		n/A	3.2 ± 2.6	
		Clostridium	Increase	3.1 ± 2.6		8.5 ± 4.1 ***		1.6 ± 1.6	3.5 ± 3.1	
		Erysipelothrix	Increase	n/A		1.3 ± 1.0 ***		n/A	0.42 ± 0.59	
		Coriobacterium	Decrease	0.30 ± 0.25		1.3 ± 0.75 ***		0.25 ± 0.46	0.49 ± 0.46	
**Liu et al., 2010**	Actinobacteria Firmicutes Firmicutes	Bifidobacterium * Lactobacillus * Bacillus *	Increase Increase No change	9.6 ± 1.2 5.6 ± 2.3 3.0 ± 1.9		10.8 ± 0.4 *** 7.4 ± 1.0 2.9 ± 1.3		9.7 ± 1.1 6.3 ± 1.8 2.7 ± 1.1	8.8 ± 2.4 6.0 ± 1.7 2.8 ± 1.2	PCR amplification of the V2-V3 region of the 16S rDNA gene obtained from fecal samples
	Proteobacteria Firmicutes	Pseudomonas ** Enterococcus **	Decrease No change	2.6 ± 1.5 9.8 ± 1.2		2.1 ± 0.4 *** 10.5 ± 0.7 ***		2.5 ± 1.2 10.4 ± 0.7	2.7 ± 1.3 10.5 ± 0.5	
	Bacteroidetes Firmicutes Firmicutes	n/A n/A Staphylococcus	Increase Decrease No change	7.9 ± 1.5 7.6 ± 1.1 3.8 ± 1.5		8.9 ± 0.7 6.4 ± 1.2 3.6 ± 1.0		8.0 ± 1.3 7.5 ± 1.0 3.5 ± 1.3	8.7 ± 1.1 8.3 ± 1.0 3.5 ± 1.2	
**Gianotti et al., 2010**	Proteobacteria	n/A	Decrease	Low dose	High dose	Low dose	High dose	n/A	4.5 ± 0.2	PCR amplification, material was obtained from fecal samples
				n/A	n/A	4.6 ± 0.6	2.4 ± 0.3 ***			
	Firmicutes	Enterococcus **	Decrease	n/A	n/A	4.1 ± 0.4	3.4 ± 0.7 ***	n/A	4.3 ± 0.5	
**Rafter et al., 2007**	Actinobacteria Firmicutes	Bifidobacterium * Lactobacillus *	Increase Increase	7.52 ± 1.58 5.68 ± 1.51		8.76 ± 0.90 *** 6.79 ± 1.39 ***		7.67 ± 0.93 7.39 ± 0.89	8.08 ± 0.96 7.74 ± 1.38	standard plate count techniques, material obtained from fecal samples
	Firmicuites	Enterococcus **	No change	5.26 ± 1.02		6.44 ± 1.13		6.74 ±1.07	6.11 ± 1.23	
	Firmicuites Coliforms Bacteroidetes	Clostridium n/A Bacteroides	Decrease Decrease No change	4.01 ± 2.22 5.33 ± 1.37 7.12 ± 1.10		3.79 ± 2.69 *** 5.63 ± 1.35 7.24 ± 1.34		3.90 ± 2.49 6.08 ± 1.19 7.47 ± 1.10	3.03 ± 2.30 6.10 ± 1.00 7.92 ± 1.32	

Mizuta et al. The data for the microbiota were expressed as the median and interquartile range (IQR) of the proportion of each bacteria, * potentially beneficial bacteria, ** potentially harmful bacteria; *** *p* < 0.05.

## Data Availability

To get an access to secondary data please contact correspondence author.

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
