# Peer review of "Role of Probiotics in Modulating Human Gut Microbiota Populations and Activities in Patients with Colorectal Cancer—A Systematic Review of Clinical Trials"

_nutrients, 2021, doi:10.3390/nu13041160_

Round 1
Reviewer 1 Report
Good review on this emerging field topic although there are several recent reviews on a similar theme. Aim of the review is unclear - is it to assess changes in gut microbiota following probiotics or is it clinical outcomes. Methodology seems appropriate, but search through databases isn't extensive (for example no EMBASE or Cochrane). Unclear why abstracts were excluded as they can be a useful resource for as yet unpublished RCTs. Also a good idea to submit protocol to PROSPERO and present their detailed search strategy as a supplement. Would refrain from using the word dysbiosis in the table and in text as it does not mean much when comparing two groups. For example when the authors say dysbiosis reduces, it is not clear what this denotes. They should rather explicitly state significant changes in alpha / beta diversity / and specific taxa. Table 3 is not helpful and has the potential to be very misleading. Mean changes as defined by absolute values on specific taxa has no relevance if they are not significant (based on p values or confidence intervals) and are not corrected for multiple testing hypothesis. Also stating certain bacteria as "harmful" or "beneficial" are meaningless without explaining why in the context of colorectal cancer. Furthermore sample sizes for these studies are very small leading to type 1 errors. Adequate critique of these studies are missing (including compliance with treatment, bias, randomisation, follow up/attrition periods etc - using cochrane risk of bias / CASP tools for eg) As the discussion and results are based on these findings, may not be accurate.Author Response
We would like to thank the reviewer for her/his thoughtful comments and efforts towards improving our manuscript. The point-to-point response was included below:
1.Good review on this emerging field topic although there are several recent reviews on a similar theme. Aim of the review is unclear - is it to assess changes in gut microbiota following probiotics or is it clinical outcomes.
Thank you for this comment. The aim of this review was to assess changes in gut microbiota following probiotic administration. Additionally, the changes in i.e. postoperative improvement, or in inflammatory markers were presented in our systematic review.
- Methodology seems appropriate, but search through databases isn't extensive (for example no EMBASE or Cochrane). Unclear why abstracts were excluded as they can be a useful resource for as yet unpublished RCTs. Also a good idea to submit protocol to PROSPERO and present their detailed search strategy as a supplement.
Thank you for this comment. As Cochrane Reviews and Editorials are indexed in PubMed and other databases, including Web of Science and Scopus we did not look at this database separately. Unfortunately, we do not have access for EMBASE. Abstract usually will not give full-information on paper and in general, there is not so easy to get in touch with corresponding author to get additional information via e-mail. Of course, the PROSPERO is known for us and we have used it for different reviews to get first overview on selected topic. Nevertheless, it should be done before publication, currently it is to late.
- Would refrain from using the word dysbiosis in the table and in text as it does not mean much when comparing two groups. For example when the authors say dysbiosis reduces, it is not clear what this denotes. They should rather explicitly state significant changes in alpha / beta diversity / and specific taxa.
Thank you for this valuable comment, we are sorry for the confusion. Now, we have clarified how dysbiosis was measured, by writing alpha and beta diversity in adequate studies.
- Table 3 is not helpful and has the potential to be very misleading. Mean changes as defined by absolute values on specific taxa has no relevance if they are not significant (based on p values or confidence intervals) and are not corrected for multiple testing hypothesis.
Thank you for this comment. Most of the analyzed bacteria changed significantly during probiotics supplementation, which was underlined by „*” in the table.
- Also stating certain bacteria as "harmful" or "beneficial" are meaningless without explaining why in the context of colorectal cancer.
Thank you for this comment, we have added information on “potentially harmful or potentially beneficial” bacteria, (the discussion section), explaining their effect in regards to colorectal cancer.
- Furthermore sample sizes for these studies are very small leading to type 1 errors. Adequate critique of these studies are missing (including compliance with treatment, bias, randomisation, follow up/attrition periods etc - using cochrane risk of bias / CASP tools for eg) As the discussion and results are based on these findings, may not be accurate.
Thank you for this comment. Due to the high heterogeneity of the included study, the meta-analysis was not performed. In the limitation of the study we added specific critique of the included studies.
Moreover, the risk of bias with the use of Cochrane tool was added, which confirmed accuracy of the presented data.
Reviewer 2 Report
Overview: This systematic review aimed to identify recent studies of the use of probiotics and synbiotics as adjuvant treatment in colorectal cancer. The authors used appropriate systematic review schema and included 6 studies in the final analysis. They identified the species and duration of probiotics and prebiotic used for each study and noted effects on inflammatory markers and “dysbiosis”, as well as effects on other side effects due to treatment. The authors conclude that overall the use of probiotics and synbiotics have beneficial effects on the gut microbiome and inflammatory milieu.
Major points of concern:
- From my review of the literature it appears that one recent 2019 study (DOI
https://doi.org/10.1186/s12876-019-1047-4) is missing but should have been included in this analysis. Why was this study not included, it appears to meet all of your criteria.
- Please pay attention to the use of italics when writing out names of genus names and below; they should all be italicized. Some are and some are not, please be consistent.
- In Table 2, “Dysbiosis’ is an unclear term; how was “dysbiosis’ measured? Alpha diversity, beta-diversity, please be specific. How are you defining dysbiosis?
- As chemotherapy-induced diarrhea (CID) affects over half of CRC patients, it would be helpful to know if there was any information on mitigations of side-effects (e.g. CID)? or survival/mets? Please include these details to improve clinical relevance.
- In Table 3, you don’t need family level information - please remove.
- In Table 3, is this mean relative abundance? why are numbers reported differentially throughout table, please be consistent.
- In Table 3, Fusobacteria in itself is not a harmful bacterium, only when it develops virulent qualities; same with these others listed as “harmful”. Please list as “potentially harmful” or similar. Further, was 16S analysis from feces or tissue? Was diet controlled for in any of these studies? All of this information is critical for interpretation of results.
- There are several issues throughout the paper where a hyphen is used when it is not necessary, please resolve; especially in the title.
Author Response
We would like to thank the reviewer for her/his thoughtful comments and efforts towards improving our manuscript. The point-to-point response was included below:
- From my review of the literature it appears that one recent 2019 study (DOI
https://doi.org/10.1186/s12876-019-1047-4) is missing but should have been included in this analysis. Why was this study not included, it appears to meet all of your criteria.
Thank you for this valuable comment and recommendation of a publication by Zaharuddin et al. entitled “Randomized double-blind placebo-controlled trial of probiotics in post-surgical colorectal cancer”. In this interesting study, authors determined the effect of probiotic consumption containing six viable microorganisms of Lactobacillus and Bifidobacteria strains on clinical outcomes and inflammatory cytokines in patients with colorectal cancer. Unfortunately, this publication does not contain crucial for our systematic review information regarding specific changes of gut microbiota following probiotic intervention. That is why we did not include this study to our publication. However, we added this publication in the discussion section.
- Please pay attention to the use of italics when writing out names of genus names and below; they should all be italicized. Some are and some are not, please be consistent.
Thank you for this comment, we corrected the italics accordingly.
- In Table 2, “Dysbiosis’ is an unclear term; how was “dysbiosis’ measured? Alpha diversity, beta-diversity, please be specific. How are you defining dysbiosis?
Thank you for this comment, we are sorry for the confusion. Now, we have clarified how dysbiosis was measured, by writing alpha and beta diversity in adequate publications. We are defining dysbiosis as a reduction in microbial diversity and the increase in proinflammatory species. This imbalanced microbiota is not able to protect from pathogenic organisms, and further can trigger inflammatory conditions and produce carcinogenic or genotoxic metabolites.
- As chemotherapy-induced diarrhea (CID) affects over half of CRC patients, it would be helpful to know if there was any information on mitigations of side-effects (e.g. CID)? or survival/mets? Please include these details to improve clinical relevance.
Thank you for this valuable comment. Unfortunately, the included studies in our systematic review did not have any information on mitigations of side-effects or survival in patients with chemotherapy-induced diarrhea. Only one study (Mizuta et al.) enrolled patients with preoperative chemotherapy. In the remaining studies patients underwent surgical procedures. Unfortunately, the contact with corresponding authors were impossible.
- In Table 3, you don’t need family level information - please remove.
Thank you for this comment, we removed the family level information accordingly.
- In Table 3, is this mean relative abundance? why are numbers reported differentially throughout table, please be consistent.
Thank you for this comment. Changes in gut microbiota were presented in 4 out of 6 studies as mean±SD. One study as % and one as inter-quartile range…To be more informative we have added the significance to each of them to show the meaningfulness of the presented changes.
- In Table 3, Fusobacteria in itself is not a harmful bacterium, only when it develops virulent qualities; same with these others listed as “harmful”. Please list as “potentially harmful” or similar. Further, was 16S analysis from feces or tissue? Was diet controlled for in any of these studies? All of this information is critical for interpretation of results.
Thank you for this comment, we corrected the listed bacteria into “potentially harmful”. Furthermore, we explained their effect in regards to colorectal cancer.
We included the detailed information regarding 16S sequencing in the Table 3. All studied used the fecal samples, additionally only one publication harnessed mucosal and tumor tissue samples. In order to unify presented and further compared data contained in the table, we used only the information regarding microbiota changes based on fecal samples.
Unfortunately, only one publication mentioned about the diet (Liu et al.), where patients received regular diet preoperatively and a low residue diet 1 day before surgery. This information was added in the results section. The remaining publications do not have information regarding this matter.
- There are several issues throughout the paper where a hyphen is used when it is not necessary, please resolve; especially in the title.
Thank you for this comment, we removed the hyphen where it was not necessary.
Round 2
Reviewer 1 Report
The authors have made most of the changes to the manuscript. Two minor concerns the lack of access to databases for a comprehensive search may not make this as systematic as expected. Also a critique of the limitations of these studies are missing and the authors should include this and consider softening the conclusions
Author Response
We would like to thank the reviewer for her/his thoughtful comments and efforts towards improving our manuscript.
The point-to-point response was included below:
The authors have made most of the changes to the manuscript. Two minor concerns the lack of access to databases for a comprehensive search may not make this as systematic as expected. Also a critique of the limitations of these studies are missing and the authors should include this and consider softening the conclusions
Thank you for this comment. The critique of these studies was added and conclusions were softened.
Reviewer 2 Report
The authors adequately addressed all of my comments. Just one minor issues with with the sentence on line 188. It should say "nevertheless" instead of "in any case".
Author Response
We would like to thank the reviewer for her/his thoughtful comments and efforts towards improving our manuscript.
The point-to-point response was included below:
The authors adequately addressed all of my comments. Just one minor issues with with the sentence on line 188. It should say "nevertheless" instead of "in any case".
Thank you for this comment, it was changed.